# The Effect of Plyometric Training in Volleyball Players: A Systematic Review

**DOI:** 10.3390/ijerph16162960

**Published:** 2019-08-17

**Authors:** Ana Filipa Silva, Filipe Manuel Clemente, Ricardo Lima, Pantelis T. Nikolaidis, Thomas Rosemann, Beat Knechtle

**Affiliations:** 1Polytechnic Institute of Viana do Castelo, School of Sport and Leisure, 4960-320 Melgaço and Research Nucleos of Polytechnic Institute of Maia (N2i), 4475-690 Maia, Portugal; 2Polytechnic Institute of Viana do Castelo, School of Sport and Leisure, Melgaço and Instituto de Telecomunicações, Delegação da Covilhã, 6200-001 Covilhã, Portugal; 3Polytechnic Institute of Viana do Castelo, School of Sport and Leisure, Melgaço, CIDESD—The Research Center in Sports Sciences, Health Sciences and Human Development, 4960-320 Melgaço, Portugal; 4Exercise Physiology Laboratory, Nikaia 18450, Greece; 5Institute of Primary Care, University of Zurich, Zurich 8091, Switzerland; 6Medbase St. Gallen Am Vadianplatz, St. Gallen 9001, Switzerland

**Keywords:** plyometrics, performance, jump, strength, flexibility, agility

## Abstract

Volleyball is considered a very explosive and fast-paced sport in which plyometric training is widely used. Our purpose was to review the effects of plyometric training on volleyball players’ performance. A systematic search was conducted according to the preferred reporting items for systematic reviews and meta-analyses (PRISMA) guidelines using PubMed, SciELO, SPORTDiscus, Medline, Scopus, Academic Search Complete, CINAHL and Web Science for articles published no later than December 2018. Any criteria were imposed for the included sample. The search focus was on interventional studies in which athletes underwent a plyometric program. To the 1831 articles found, another five were added, identified through other sources. Duplicated files were removed, titles and abstracts were screened, which left 21 remaining studies for extensive analysis. Results showed that the vertical jump (15 studies) was the major ability studied in plyometric training interventions, followed by strength (four studies), horizontal jump (four studies), flexibility (four studies) and agility/speed (three studies). In addition, it was observed that young (under 18 years old) female athletes were the most studied. The included studies indicated that plyometric training seems to increase vertical jump performance, strength, horizontal jump performance, flexibility and agility/speed in volleyball players. However, more studies are needed to better understand the benefits of plyometric training in volleyball players’ performance.

## 1. Introduction

Volleyball is an intense anaerobic sport that combines explosive movements (i.e., in both vertical and in horizontal directions) with short periods of recovery [1,2,3]. Therefore, explosive strength, which is defined as the ability of an individual’s neuro-muscular system to manifest strain in the shortest possible time [4], is considered a fundamental aspect of successful athletic performance (e.g., [5]). In fact, when speed and agility are combined with maximum strength, power is the outcome [6]. Muscular power enables a given muscle to produce the same amount of work in less time, or a greater magnitude of work in the same time, which is important for sprinting, jumping [7] and quick changes of direction [6]. Indeed, studies have shown strong relationships between power measures and vertical jump performance (e.g., [7,8,9]), suggesting that power influences vertical jumping performance [10].

A vertical jump is a complex movement that requires the coordination of several muscles in the trunk, arms and legs [11]. Knowing that each player performs more than 250 jumps in a volleyball match of five sets [12,13], jumping ability has been identified as one of the key determining factors of high performance in volleyball [14].

In fact, several studies have shown that vertical jump test results are indicative of the performance level of an athlete (e.g., [15,16,17,18]). For example, Smith [17] found that vertical jump performance during spiking and blocking was greater in Canadian national volleyball players compared to the Canadian university volleyball players. Also, Ziv and Lidor [16], in a review concerning vertical jump in female and male volleyball players, noted that better-performing teams were comprised of players with high vertical jumps [19].

Jump training is commonly associated with plyometric training and, in particular, with drills that stress the musculotendinous unit [20,21,22]. In fact, de Villarreal [23] found that a combination of bodyweight plyometrics, including countermovement jumps, depth jumps and squat jumps, resulted in a 4.7% to 15% increase in vertical jump height. Nevertheless, this type of training increases neuromuscular coordination through training the nervous system [24], thus allowing the stretch-shortening cycle (SSC)—which is a lengthening movement (i.e., eccentric) quickly followed by a shortening movement (i.e., concentric) [20,24]—to react faster [25]. Additionally, because this training includes muscle lengthening, it may also improve flexibility, increase the amount of stored elastic energy in the muscles [26], stimulate more muscle units [27], result in higher (neural) firing frequency [27,28] and improve joint proprioception [29,30].

According to the concept of training specificity, the effective transfer of training adaptations occurs when the training exercises match the task (e.g., testing, competition; [31]). In volleyball, plyometric training involves jumping, hopping and bounding exercises as well as throws that are performed quickly and explosively [32,33]. Those movements are also related to the development of agility [34,35,36]. This capability is thought to be a reinforcement of motor programming through neuromuscular conditioning and the neural adaptation of muscle spindles, Golgi tendon organs and joint proprioceptors [34].

In addition, the athlete’s age and sex should be considered when planning strength training programs. For instance, during adolescence, the changes that occur in muscular, neuronal and hormonal systems [37] due to the development related to puberty (namely the growth spurt) influence adolescents’ abilities to execute movements [38]. Moreover, the female growth spurt occurs approximately two years earlier than the male growth spurt and reaches its plateau at 15–16 years of age, whereas males continue their growth spurt up to the age of 19–20 years [39]. Due to these changes in adolescence, it was found that female athletes had weaker quadriceps and hamstring muscles (even when normalized for body weight) in the adult stage when comparing to their male counterparts [40]. Those dissimilarities reflect the different ability to produce strength, influencing jumping performance and is reflected in the different motor pattern showed by the two sexes [40].

Although plyometric training has been widely used in volleyball, little scientific information is available to determine its possible benefits on the different components of performance. Following this, the aim of this systematic review was twofold: (i) To evaluate the efficacy of plyometric training programs on both male and female volleyball players, and (ii) to understand the effect of those programs based on players’ age.

## 2. Materials and Methods

### 2.1. Literature Search Strategy

A systematic literature search was carried out in PubMed, SPORTDiscus, MEDLINE, Scopus, CINAHL Plus, Academic Search Complete, Web of Science and SciELO. The search was limited to original articles written in English published online no later than December 2018, which is when the search was conducted.

The words used in databases combined terms related to plyometric training (strength training OR power training OR plyometric training OR resistance training OR weight training OR complex training OR weight-bearing exercise OR eccentric training) and volleyball.

### 2.2. Selection Criteria

The process for screening articles followed the preferred reporting items for systematic reviews and meta-analyses (PRISMA) guidelines [41]. The inclusion criterion included interventional studies which provided a plyometric strength program in volleyball team sports and which used statistical analyses to quantify the association between the training implemented and its benefits regarding performance. No restrictions were imposed on the participant sample in terms of age, sex, playing level, playing experience, etc. Literature reviews, overviews, conference proceedings and both masters and Ph.D. thesis were excluded.

The abstracts of all the articles found were screened against the predefined selection criteria by the authors of the present study. Cases of disagreement were discussed among the researchers until a consensus was established. The same process was used to screen the full-text version of each article. Furthermore, a few relevant articles that were not found in the first literature search—possibly due to discrepancies in the terminology used to describe plyometric training—were included.

A risk of a bias within the studies was independently assessed by the two reviewers considering the topics of (i) a bias in selecting participants into the study; (ii) a bias in classifying interventions; (iii) a bias due to departures from intended interventions; (iv) a bias from missing data; (v) a bias in measurement of outcomes; and (vi) a bias in reporting outcomes selectively. Studies with a risk of bias in all categories were excluded from the review. Agreement between reviewers was tested using the Kappa index test that revealed a value of 0.89, suggesting a very good agreement.

### 2.3. Assessment of Methodological Quality

Methodological quality was assessed using the STROBE Statement, which is a 22-item checklist considered essential for the accurate reporting of observational studies. This checklist includes a link between the title of the article and its abstract (item 1), introduction (items 2 and 3), methods (items 4 to 12), results (items 13 to 17) and discussion (items 18 to 21) sections, and any other information (item 22). From those, 18 items are common to all three designs, while four (items 6, 12, 14 and 15) are design-specific, with different versions for all or part of the item. For some items (indicated by asterisks), the information should be given separately for cases and controls in case-control studies, or for exposed and unexposed groups in cohort and cross-sectional studies. Each article was classified based on the sum of the points for all 22 items (one point was counted for an item if the criteria was achieved), and the result was divided by the maximum possible point total of 22 (e.g., if an article had 11 points, the value arrived at was 0.5).

The items of all articles were independently classified by each of the observers, and then an interobserver reliability analysis was conducted. The Kappa index test revealed a value of 0.94 (90% IC: 0.92–0.96), indicating very good agreement between observers.

### 2.4. Data Extraction and Analysis

For the articles included in this study, all authors discussed how the information should be organized regarding the characteristics of the studies and the results of the assessed measurement properties. Afterwards, two independent reviewers extracted data regarding the participants’ characteristics (i.e., number, age and skill level) and the results of the implementation of the plyometric training on the physical component(s) studied in each article (i.e., vertical jump, agility/velocity, strength, power, horizontal jump and flexibility).

## 3. Results

### 3.1. Search, Selection and Inclusion of Publications

From the database search, 1381 files were found to which five more were added from other sources. These data were then exported to reference manager software (EndNote^TM^ X8, Clarivate Analytics, Philadelphia, PA, USA), and 314 duplicates were eliminated. The remaining 1072 files were screened for relevance according to their titles and abstracts; through this process, another 1016 articles were removed. After the full texts of the remaining 56 articles were read, another 24 were excluded due to a lack of relevance to the specific purpose of the current study. In the final step of the screening procedure, another 13 articles were eliminated either because they included other sports in their analysis (*n* = 2), added another strength methodology training (*n* = 9), studied the effect of training on postural control (*n* = 1) or indicated no experimental intervention (*n* = 1). After the entire screening process was complete, 19 files were deemed acceptable for inclusion in the present review. A schematic summary of this search is shown flowchart in Figure 1.

### 3.2. Studies Score

Studies were evaluated through the STROBE scale and Table 1 presents the score achieved by each study.

### 3.3. Data Organization

The data for the present study were grouped according to the effects of plyometric training on the different physical fitness components studied in volleyball players (i.e., vertical jump, strength, horizontal jump, flexibility and agility/speed). Each of the three authors of the present study independently classified the papers according to the different research topics (components). Disagreements were resolved through discussion among the three authors until a consensus was reached.

#### 3.3.1. Effects of Plyometric Training on Vertical Jump Performance

Fourteen of the 19 articles included in this systematic review offered inferences about the effects of plyometric training on vertical jump performance (Table 2). The vertical jump tests used in these studies included the squat jump, countermovement jump, drop jump, standing vertical jump, single leg jump and repeated jumps (15 and 30 s). Eleven of the 14 studies which reported data about the effects of plyometric training on vertical jump performance were exclusively conducted on women ranging from 14 to 22 years of age, most of which (*n* = 9) were conducted in under-18 women players. No study tested the effects of plyometric training exclusively in men; two of the articles analyzed the effects of plyometric training in both men and women.

The training protocols varied between two and three training sessions per week, and the total intervention period ranged from four to 16 weeks. The most common protocols lasted 6 (*n* = 4) or 12 (*n* = 4) weeks. In all the studies, improvements in vertical jump performance were observed after the intervention. The plyometric training protocols varied, being carried out either on a traditional gymnasium floor, grass or concrete, while some studies included aquatic plyometric sessions. However, the most common setting was a traditional gymnasium floor. Frequently, training programs included both leg jumps, single-leg jumps, hurdle jumps, drop jumps, box jumps or lunge jumps. Ball throws were the most common upper-body exercise performed.

After a four-week-long aquatic plyometric training intervention conducted on under-14 women players, an improvement of 3.9 cm in the players’ vertical jumps was seen [42]. The vertical jump measured for under-15 players also improved (by 9.2%) after six weeks of regular plyometric training on a gymnasium floor [44]. After a plyometric training protocol that lasted 12 weeks and which was conducted in under-17 women players, improvements of 16.9% in counter-movement jumps were found [46]. Also, for the same test in under-22 women players, an improvement of 27.6% was recorded after 12 weeks of plyometric training [48]. In a study conducted in both men and women players (~21 years old), it was observed that plyometrics training interventions carried out on grass and on concrete for four weeks promoted improvements of 3.34 cm and 3.67 cm, respectively, in the counter-jump performance of players [50].

#### 3.3.2. Effects of Plyometric Training on Strength Performance

Four of the 19 articles included in this systematic review presented inferences about the effects of plyometric training on the strength performance of athletes (Table 3). The strength tests included those for peak torque of lower limb (i.e., concentric and eccentric peak torque), isokinetic peak torque of hamstrings and weight and plyometric training. The literature included two studies with male volleyball players and two with female volleyball players, ranging from 14 to 21 years of age, with three of these studies being conducted in youth. No studies compared men’s and women’s results. All four studies are cohort-studies which included either a six-week intervention period [42,44,52] or a 12-week intervention period [54].

Beyond the influence of plyometric training on the stiffness of the lower limbs [52], research has highlighted the effect of aquatic plyometric training on strength performance, which improves the concentric peak torque in the dominant leg [42]. Furthermore, weight training sessions that include exercises for the upper body, lower body, and trunk increase isokinetic peak torque by 13% on the dominant side and by 26% on the non-dominant side [44].

#### 3.3.3. Effects of Plyometric Training on Horizontal Jump

Four studies included in this systematic review presented inferences about the effects of plyometric training on horizontal jump performance (Table 4). The horizontal jump tests included two exercises: The standing long jump and the depth leap long jump. One study included both male and female participants [56], one included only males [58] and another included only females [48]. The study conducted by Milić et al. [52] did not report the genre of the participants. Two studies are cohort studies involving competitive-level youth players (one of which employed a six-week intervention period and one of which utilized a 16-week intervention period), and the other two studies were randomized controlled studies involving 12-week intervention periods with athletes between the ages of 18 and 24 years.

Studies conducted by Milić et al. [52] and Çımenlı et al. [57] revealed a significant increase in horizontal jump performance after the plyometric training intervention. However, another study did not show significant differences in this regard [56], and the study carried out by Gjinovci et al. [48] presented only a small effect of plyometric training on horizontal jump performance.

#### 3.3.4. Effects of Plyometric Training on Flexibility

Only two articles included in this systematic review presented inferences about the effects of plyometric training on flexibility (Table 5). The flexibility measure was based on the sit and reach test, which is used to examine the flexibility of the hamstring and lower back. The participants of these two studies were female youth volleyball players. One study was a cohort study with a 12-week intervention period [43], and the other was a randomized controlled trial which also had an intervention period of 12 weeks [45]. The results of these studies reveal that plyometric training can improve flexibility by 9% [43] to 14% [45].

#### 3.3.5. Effects of Plyometric Training on Agility/Speed

This systematic review found three studies which observed the effects of plyometric training on the agility/speed of volleyball players [43,46,47]. The results can be found in Table 6. The tests used in these studies included a 20 m sprint [43], a zigzag double-leg jump over the line combined with a shuttle run (6 m × 6 m) [46], and a shuttle run for which the distance was not reported [47].

Two of the three studies included female youth volleyball players; the other study included 18- to 22-year-old male volleyball players.

The results of one study revealed that an eight-week intervention period significantly improved participants’ speed [46]. Corroborating these results, a post hoc analysis indicated improvements in speed values in a 20 m sprint [43]. Finally, after a 12-week plyometric training program, it was shown that participants who completed the training program while wearing a weighted vest showed greater improvements in agility than those who did not wear a weighted vest [47].

## 4. Discussion

### 4.1. Effect of Plyometric Training on Vertical Jump Performance

The purpose of plyometric training is to improve the power of subsequent movements using both the natural elastic components of the muscles and tendons as well as stretch reflex [49]. Considering that jump performance ability is highly influenced by the individual’s ability to take advantage of the elastic and neural benefits of the SSC, well-developed strength and the rate of excursion of the activated musculature during the contraction, it is expected that plyometric training may benefit athletes’ jumping performance [51]. In fact, the literature is consistent in suggesting that plyometric training contributes to the optimization of landing mechanisms [53], improvements in eccentric muscle control and an increase in knee flexion and hamstring activity [55].

Improvements in jumping performance (independent of the type of jump analyzed) were observed in all the studies presented in Table 2. Moreover, such benefits were observed in both males and females and independently of age. In theory, it is expected that meaningful improvements on the jumping performance will be observed after the implementation of a plyometric training intervention, with larger increases in counter-movement or drop jumps than in squat jumps.

Such a hypothesis is based on the fact that counter-movement and drop jumps are more dependent on SSC than squat jumps. When performing a squat jump, a pause occurs during the amortization phase, leading to the dissipation of elastic potential energy and a decrease in the potentization effect based on SSC [57]. In a study that compared plyometric training (i.e., young adults; men and women) carried out on grass with plyometric training carried out on concrete, improvements of 3.17 and 2.17 cm, respectively, in squat jump performance and of 3.34 and 3.67 cm, respectively, in counter-movement were observed [50]. However, in a study which involved a 12-week training period on young adult men and which tested more than one type of jump, it was observed that improvements were slightly greater for the squat jump (+5.93 cm) than counter-movement (+4.98 cm) and drop jumps (+4.83 cm) [59].

These two studies are not enough to generalize the idea that plyometric training is more beneficial for counter-movement and drop jumps in volleyball players. However, it is important to note that such a hypothesis has been supported in a systematic review of women athletes from different sports [51]. For that reason, future studies conducted on volleyball players should include more than one type of vertical jump test to analyze the different effects of plyometric training on players’ performance.

Throughout the articles reviewed presently, a clear tendency to conduct studies in young (under-18) women players was observed. No study included in the review analyzed men exclusively, and only a couple of articles tested the effects of plyometric training on adults. This is a limitation to understanding the general effects of plyometric training on volleyball players. In fact, the effectiveness of plyometric training may depend on different factors such as age, maturation, sex or training level [51]. Among these variables, maturation seems especially likely to be a determinant factor in explaining the effectiveness and the responsiveness of athletes to plyometric training [61]. However, in in the articles reviewed, no study analyzed the maturation status of players. This is an important gap in the research that should be considered and corrected in further studies conducted in youth players.

The effectiveness of plyometric training may also depend on the training design and the length of the intervention period [51]. Regarding the training design, it is common for researchers to describe certain variables, such as the type of exercises used, the training duration, recovery status and the frequency, volume and intensity of the training. However, there is a lack of information about other potentially important concurrent factors that may explain some of the variations in the results. In fact, the jumping technique and the influence of movement amplitude or ground-contact time were not described in any of the reviewed studies’ protocols [57].

Another common omission of information observed throughout the included studies was the type of surface that the tests were conducted on; this factor may affect the outcomes of a training program. However, in a study that tested the effects of plyometric training completed on grass or on concrete, it was observed that both training types improved the jumping performance of volleyball athletes, with no significant differences found between the two groups [50]. Also, when testing plyometric interventions in a non-solid aquatic environment, meaningful improvements in vertical jump performance were observed [42].

Another important factor that may determine the effectiveness or the amplitude of the benefits of plyometric interventions is the duration of the training period. It was observed that the interventions implemented in the reviewed studies ranged from four to 16 weeks [56,62], with periods of 6 [42,44,60,63] and 12 weeks [43,45,48,59] being the most common. In the reviewed studies, improvements of 8% [42] and 9.2% [44] in the vertical jump were reported in two of the studies that used six-week plyometric training protocols. Meanwhile, improvements of 16.9% [43] and 27.6% [48] were observed in counter-movement jumps in two of the studies which included 12-week training period protocols. It is possible that plyometric training programs of longer than 10 weeks are more helpful in obtaining meaningful improvements [51,64].

One of the main study limitations is associated with the non-report of individualized training intensity for each participant and possible confounding effects on the final outcomes. Moreover, it is necessary for a deep understanding of the effects of plyometric training on the reactive strength index.

### 4.2. Effects of Plyometric Training on Strength

Volleyball is a complex sport for which several athletic demands need to be developed [65]. One of these demands is strength because of its relevance to the technical skills used in volleyball, such as jumping, hitting and blocking. Thus, muscular strength is one of the most important factors that give players an advantage during elite-level competitions [66].

The effect of plyometric training on strength was analyzed in four studies [52]. The strength tests included examinations of peak torque of lower limb (concentric and eccentric peak torque) [42,44], isokinetic peak torque of hamstrings [52] and a combination of weight and plyometric training [54]. The benefits of plyometric training in the muscle strength was observed in both sexes and in all ages analyzed (i.e., 14–21 years old).

Beyond the influence of plyometric training on the stiffness of lower limbs [52], one of the analyzed studies showed that plyometric training improved strength performance in the dominant leg of participants of under-15 women [42]. Also, strength performance was improved when weight and plyometric training were combined and included exercises for the upper body, lower body and trunk in under-17 men and women [54]. Furthermore, plyometric training was found to improve strength on both the dominant side (26%) and non-dominant side (13%) in terms of isokinetic peak torque in under-15 women [44]. The results revealed that plyometric training increases strength regardless of the number of weeks spent training or the assessment procedures used. However, contrary to the results found in the review, in a study that assessed the level of specific lower limb power and reactive force in young female volleyball players, the stiffness test revealed no significant differences for any variables after 10 weeks of plyometric training [67].

Some possible explanations to justify the gains of strength are that plyometric training requires an appropriate technical ability as well as sufficient levels of muscle strength and joint coordination, thus increasing the inter- and intra-muscle capacity to contract and produce force [68,69]. However, it seems that a combined training program may also contribute to benefits in muscle strength.

When comparing the impact of short-term training with resistance plus plyometric training (RT + P) or electromyostimulation plus plyometric training (EMS + P) on explosive strength production in elite volleyball players, the results indicate that the first is effective in promoting jump performance, while the latter helps with increasing the jump performance, speed and agility of elite volleyball players [70]. Besides that, one study showed that an eight-week-long training program which combined jumping and ball throwing training resulted in significantly improved muscular performance in young female volleyball players [45].

One possible explanation for the results reported in three of the four studies in this review which analyzed six-weeks plyometric training programs in youth athletes is that, as mentioned in the research of Ziv and Lidor [16], at least eight weeks of training are needed, specifically for motor capacity, for the development of strength, especially when the participants are young.

Beyond the few studies reported, there is a lack of evidence about the plyometric effect on essential strength parameters such as force production rate in both concentric and eccentric phases. The improvement of future studies should consider such an analysis.

### 4.3. Effects of Plyometric Training on Horizontal Jump

The effects of plyometric training on horizontal jump performance were analyzed in four studies [48,56,58,60] that included both sexes and ages between 14 and 24 years old. The benefits of plyometric on horizontal jump were observed in both sexes and across the ages. Standing long jump [48,56,60], depth leap long jump [56], triple standing jump [60] and unilateral jumps with either no steps or one step taken [58] were used as tests. In the standing long jump, meaningful improvements of 7.6% were observed in senior female players after 12 weeks of plyometric training [48], a 7.6% improvement was observed in under-16 players after six weeks of training [60], and a 3.6% improvement was seen in 12- to 19-year-old players after 16 weeks of training [56].

These findings suggest that plyometric training positively affects horizontal jump performance, albeit with improvements lower than those recorded for vertical jump performance (9% to 28%, as observed previously). One possible explanation for the weaker effect of plyometrics on horizontal jump performance in comparison to vertical jump performance is the specificity of the plyometric training and the optimization of the force vector and muscle stimulation during the exercises. Moreover, horizontal jumping requires both vertical and horizontal actions, and so the increased complexity of the technique may be responsible for the smaller effects of plyometric training. Still, further studies should analyze the mechanisms that lead to horizontal jump improvements, and the plyometric training program should possibly include more horizontal jumping exercises to optimize this capacity. It is possible to hypothesize that the specificity of plyometric training should be considered in the training effects, namely, to improve the direction of the forces and to translate the benefits for the field.

The type of horizontal jump should also consider since the main effect of plyometric training is to increase the SSC. If the horizontal jump does not require a countermovement action, it is expectable that the effects of plyometric training will not be so pronounced.

### 4.4. Effects of Plyometric Training on Flexibility

Flexibility is a specific action for a given joint and may have different results according to the sport [71]. The sit and reach test was used to measure the flexibility of the hip and back flexion as well as lower limb extension (hamstring) [56]. In both studies that investigated flexibility [43,56], the results revealed that plyometric training improves flexibility in under-16 women. Although few studies have assessed the impact of flexibility on volleyball players and its positive effect on vertical jump performance, the findings are ambiguous when compared to findings in studies involving other sports. One the one hand, hamstring flexibility (as measured by knee extension angle) is associated with a decrease in vertical jump height in high school students [72], while, on the other hand, flexibility is reported to be a key beneficial factor in sprinting, jumping, agility and kicking in youth football players [73].

Furthermore, in a study that compared the individual and combined effects of a plyometric training program and dynamic stretching on muscular strength endurance and flexibility in 45 female collegiate volleyball players, it was shown that plyometric exercises improve several functions of the nervous system and that dynamic stretches increase muscle temperature, stimulate the nervous system and improve muscle elasticity, thus increasing flexibility by 10.29% [74]. Corroborating with the previous study, Ozgul [75] found that static, dynamic and PNF flexibility exercises improve the vertical jump performance of basketball and volleyball players.

Thus, despite the findings reported in previous studies, plyometric training may be an effective way to increase athletes’ flexibility, which could facilitate improvements in jump performance, agility and speed. The benefits of plyometric training can be explained by the activation of SSC that requires stretching with a contraction of the muscle, thus possibly justifying the benefits of eccentric component [76].

Future studies should consider the range of movements during plyometric training, aiming to determine which parameters may contribute to an increase in the flexibility and mobility of participants. Moreover, it is necessary for more comparative studies to be conducted with control groups aiming to determine the real benefits. Finally, an analysis of the length of fasciculus is important.

### 4.5. Effects of Plyometric Training on Agility/Speed

Few studies have considered the possibility that the agility and sprinting performance of volleyball players can be improved through plyometric training [43,46,47]. However, theoretically, plyometric training may help volleyball players develop both capacities. Sprint performance requires an explosive concentric and SSC force production in the lower limb muscles and can be benefited notably by the ability of players to use and optimize the elastic and neural properties of the SSC after plyometric training [77]. Also, agility, which is multifactorial and very complex, may be bolstered by plyometric training involving different neuromuscular adaptations (e.g., increased intermuscular coordination and firing frequencies), leading to a greater rate of force development and power output [64,75].

Two studies tested the benefits of plyometric training on the agility of volleyball players in both sexes. In a study conducted over eight weeks in under-15 women, it was observed that performance at in a shuttle run (6 m × 6 m) was significantly improved (by 0.7 s) [46]. In a longer intervention (12 weeks) conducted in young adult men, it was also found that participants’ agility in a 50 m shuttle run was meaningfully improved, as the times to complete the shuttle run decreased from 14.15 to 12.86 s for participants who wore a weighted vest and from 14.51 to 13.97 s for participants who did not wear a weighted vest [59]. Both studies confirmed the notion that plyometric training increases agility.

Plyometric training may contribute to reductions in ground contact times via increases in muscular force output and movement efficiency [78,79]. Additionally, plyometric training may improve the eccentric strength of the lower limbs, which are extremely important during the decelerations involved in short movements [78] and during the accelerations and decelerations involved in changes of direction [64]. However, such possibilities should be further researched (as should the effect of maturation) in youth players, specifically considering the relationships between maturation, the development of the central nervous system and increases in fascicle length [80].

Regarding the effect of plyometric training on sprint performance, a study involving 12 weeks of plyometric training [59] which participants completed either with or without a weighted vest showed improvements in speed during 50 m sprint tests for both groups. Average sprint times improved from 8.15 to 7.10 s for participants who wore vests and from 8.22 to 7.69 s for those who did not wear vests, whereas no significant change was observed in the control group [47]. In testing the effects of plyometric training on athletes’ performance in a shorter sprint test (20 m), improvements in the time taken to complete the test from 3.8 to 3.6 s were observed, representing very likely benefits of 5.7%.

Both studies [46,56] were promising in suggesting that improvements in the sprint performance of volleyball players occur after plyometric training. Such a capacity is possibly not the most determinant or prevalent in this sport; however, it seems that the temporal sequencing of muscle activation for more efficient movement, preferential recruitment of fastest motor units or velocity increases in nerve conduction promoted by plyometrics successfully improves sprint performance [23]. Naturally, there is a possible link between the velocity of muscle contractions during plyometric training and the transfer of energy during sprinting [81], as well as between the type of exercise and the implications for the different phases of sprinting [77]. However, the fact that plyometric training induces repeated ballistic exercises may partially explain players’ improved ability to generate explosive ground reaction forces after undergoing a plyometric training intervention [82], thus improving their acceleration during sprinting [81]. Authors should discuss the results and how they can be interpreted in the perspective of previous studies and of the working hypotheses. The findings and their implications should be discussed in the broadest context possible. Future research directions may also be highlighted.

It is important to improve future studies by adding information about the effects of plyometric training on change of direction deficits and capacity to quickly change the direction in a match.

## 5. Conclusions

The majority of the studies included (*n* = 13) focused on young players, and most of them (*n* = 12) observed only women players. However, a lack of information about the players’ maturation was noticed, which could have influenced the effectiveness of the programs. The similarities between vertical jump test and the movements conducted during the training programs seem to explain the higher number of studies (*n* = 15) and the greater improvements when comparing to horizontal jumps. Nevertheless, future studies should provide more information about the characteristics of the training programs. Studies also showed that the typical plyometric training component, the SSC, promote the necessary stimulus to improve strength, as well as flexibility. This fact is justified by coordination improvements through a great muscle unit firing, in the first, and by the required lengthening movement (eccentric), in the second. Likewise, agility/speed performance was investigated in only three studies but also seems to be improved through plyometric training, possibly due to the resultant increases in muscular force output and movement efficiency, which include the faster recruitment of the motor units and increased velocity in nerve conduction. Nevertheless, more studies should be conducted to better understand the benefits of this type of training for volleyball players’ overall performance.

## Figures and Tables

**Figure 1 ijerph-16-02960-f001:**
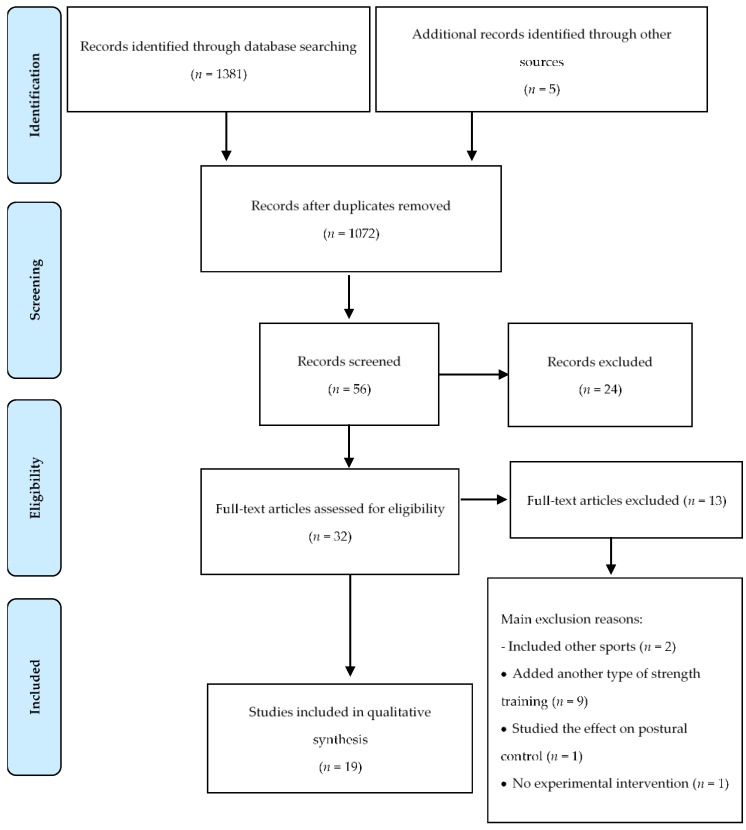
Flowchart on the literature search of plyometric training in volleyball.

**Table 1 ijerph-16-02960-t001:** Quality score for the included studies.

Study	Quality Score	Study	Quality Score
Martel, Harmer [42]	0.55	Turgut, Colakoglu [43]	0.82
Hewett, Stroupe [44]	0.45	Radu, Făgăraş [45]	0.73
Lehnert, Lamrová [46]	0.36	Hrženjak, Trajković [47]	0.32
Gjinovci, Idrizovic [48]	0.77	Trajkovic, Kristicevic [49]	0.41
Ramlan, Pitil [50]	0.41	Çankaya, Arabacı [51]	0.32
Milić, Nejić [52]	0.50	Mroczek, Superlak [53]	0.32
Vassil and Bazanovk [54]	0.27	Bashir, SulehHayyat [55]	0.41
Idrizovic, Gjinovci [56]	0.73	Çımenlı, Koç [57]	0.41
Myer, Ford [58]	0.68	Sheikh and Hassan [59]	0.55
Veličković, Bojić [60]	0.55		

**Table 2 ijerph-16-02960-t002:** Studies that investigated the effects of plyometric training programs on vertical jump performance.

Study	Sex	Age (Mean) and Competitive Level	*n*	Design	Training Protocol	Main Results
Martel, Harmer [42]	Women	Aquatic group: 14 yo Control group: 15 yo	19	Randomized controlled trial; 6-week period intervention	Aquatic plyometric training (APT): 2 × week; 45 min. Power skips, spike approaches, single- and double-leg bounding and squat jumps progressively increased from two times per session to five times per session. Bouts increased from 10 s to 30 s of maximal jump during the period. Depth jumps were performed involving three submerged boxes also progressively increasing the number of times session during the period. Control group (CG): a flexibility supervised program was conducted twice a week, consisting of three sets of 8–10 static stretches for 30 s each.	Meaningful increases in VJ were found after 4 weeks in both groups (3% in APT and 5% in CG). After 6 weeks, the APT improved 8% in comparison to the 4-week period, and no significant changes were found in control. Differences between groups revealed that players in APT jumped 1.5 cm higher than CG in baseline and 3.9 cm after the intervention period.
Lehnert, Lamrová [46]	Women	14.8 yo	11	Case reports; 8-week period intervention	2 × week The training program was divided into three cycles. The number of sets varied between 3 and 4, and the repetitions between 8 and 10. No information about the resting period was provided.	The standing VJ increased from 29.50 cm at the baseline to 30.45 cm after 4 weeks of the program and 33.54 cm at the sixth week after the completion of the program.
Milić, Nejić [52]	N.R.	16 yo	46	Case reports; 6-week period intervention	Plyometric training group: 2 to 3 times per week (15 sessions in total during the experimental period). Five exercises (hurdle jump, depth jump, box jump sideways, lunge jumps and vertical jumps) for explosive leg strength. The number of sets varied between 2 and 4, and the repetitions between 6 and 12 during the training period.	Meaningful improvements were found in the plyometric training group: The two-foot takeoff block jump improved 3.53 cm; the right foot takeoff block jump improved 3.44 cm; the left foot takeoff block jump improved 4.05 cm; the two-foot takeoff spike jump improved 5.22 cm; the right foot takeoff spike jump improved 4.34 cm; the left foot takeoff spike jump improved 5.39 cm; the standing depth jump improved 17.95 cm; and the triple standing jump improved 72 cm.
Vassil and Bazanovk [54]	Men and women	14.4 for women and 17.0 yo for men	21	Case reports; 16-week period intervention	Twice a week. Squat jumps, lateral box push-offs, overhead throws, split squats, power drop, depth jumps, lateral hurdle jumps, plyometric push-ups and single-leg lateral hops were implemented. Each session had six exercises repeated twice (two sets), varying the repetitions between 8 and 15.	VJ meaningfully improved in girls from 45.3 to 49.9 cm. Despite no significant changes being found, the jump’s height also increased in men from 62.1 to 67.2 cm.
Idrizovic, Gjinovci [56]	Women	16.6 yo	47	Randomized controlled trial; 12-week period intervention	Three groups: plyometric, skill-based and control. The plyometric and skill-based groups had two sessions per week in addition to their regular training. The plyometric training lasted 10–15 min/session, and the skill-based training lasted 20–30 min. The plyometric training consisted of upper and lower limb exercises. The sets of plyometrics per training varied between 5 and 6, and the repetitions between 1 and 5. Rest between sets varied between 2 and 5 min. The skill-based training consisted of volleyball drills, small-sided games and real-games drills.	The counter-movement jump improved 16.9% in the plyometrics group, 9% in the skill-based group, and 8.5% in the control group. *Post hoc* analysis revealed better effects of plyometrics in comparison with the other two training groups for counter-movement jump performance.
Myer, Ford [58]	Women	15.9 yo in plyometric group and 15.6 yo in balance group	18	Randomized controlled trial; 7-week period intervention	Two experimental groups: plyometric and balance. Both experimental groups participated in a common resistance training protocol. Eighteen experimental training sessions were completed. Plyometric training included (among other drills) wall jumps, squat jumps tuck jump, line jumps, lunge jumps, forward jumps and box drops. The time dedicated to each exercise varied between 10 and 20 s, and the repetitions between 3 and 10.	Plyometric training significantly increased knee flexion at the initial contact and the maximum angle in comparison to the control group during drop vertical jump tests. However, balance training increased maximum knee flexion during medial drop landing in comparison to plyometric. Both training protocols were effective in reducing lower extremity valgus measures at the hip and at the ankle and both reduced lower extremity valgus measures at the knee during a single-limb dynamic stabilization task.
Veličković, Bojić [60]	Women	14 to 16 yo	30	Case reports; 12-week period intervention	Experimental group had two sessions/week. No information about the training exercises or planification was provided.	Significant improvements in the experimental group were observed in the squat jump (+5.93 cm), counter-movement jump (+4.98 cm), drop jump (4.83 cm), and leg squat jump with preparation (+3.67 cm).
Turgut, Colakoglu [43]	Women	Weighted jump rope group: 15.0 yo; standard jump rope group 14.1 yo; control group: 14.4 yo	25	Randomized controlled trial; 12-week period intervention	Both training groups participated in three sessions/week. The control group did not participate in any training protocol. The weighted jump rope performed rope jumping with weighted ropes (600 g and 695 g). The standard jump rope consisted of a cable rope which weighed between 100 and 160 g. The training protocol for both groups varied between 30 and 60 s per repetition and between 1 and 3 sets.	The power during counter-movement jump tests was significantly improved by the weighted jump rope protocol in comparison to the control group (mean difference of 11.83 Watts). However, no meaningful differences were found between experimental groups.
Radu, Făgăraş [45]	Women	16 to 17 yo	15	Case reports; 10-week period intervention	Two plyometric sessions per week. The following exercises were included in the program: double leg and single-leg jumps; squat jumps; crossover jumps; increase and decrease jumps; broad jumps; box hop jumps; scissors jumps; single leg bounding; and power skipping.	Players meaningfully improved their overall performance at flight time, contact time, height, and power during the 15-s and 30-s jumping tests. No meaningful changes were found in stiffness.
Gjinovci, Idrizovic [48]	Women	21.9 yo	41	Randomized controlled trial; 12-week period intervention	Two experimental groups: plyometric and skill-based. Each group had two sessions per week. Plyometric training included lower-body exercises (leg hops, vertical jumps, tuck jumps, lateral/diagonal jumps, broad jumps, obstacle jumps, box jumps, and drop jumps) and upper body exercises (throwing exercises). The total of sets/week varied between 12 and 24 depending on the body part, and the repetitions between 40 and 58/week. Skill-based training consisted of volleyball drills, small-sided games, and game drills.	Both groups showed meaningful improvements in counter-movement jump performance. The plyometric group had an improvement of 27.6%, and the skill-based group had an improvement of 18%. Plyometric training was largely better than skill-based training considering the effects on counter-movement jump performance.
Hewett, Stroupe [44]	Women	15 yo	20	Case reports; 6-week period intervention	Experimental group had three sessions per week. The program followed three phases: Technique phase (2 first weeks), fundamentals phase (using a proper technique to build strength and power) and performance phase (focusing on achieving maximal jumping).	The plyometric group meaningfully improved vertical jump performance by 9.2%. Decreases in peak landing forces were observed.
Hrženjak, Trajković [47]	Women	Youth and junior plyometric group: 16.18 yo. Control group: 16.3 yo.	*N* = 60 Plyometric group (*n* = 31); control group (*n* = 29)	Randomized controlled trial; 6 -week period intervention	6 weeks; five training sessions per week (90 to 120 min). The number of training sessions was 15. The set model for development of explosive leg power consisted of five exercises, and exercises were done in the first part of the training session, after a 30-min warm-up.	Both the plyometric and the control group showed significant improvements (*p* < 0.05) in joint kinematics from pre- to post-training on most of the measures for linear velocity, except for the linear velocity in the hips during the eccentric phase (*p* = 0.669 for the plyometric group, *p* = 0.595 for the control group), where none of the group showed significant improvement.
Trajkovic, Kristicevic [49]	Women	17 yo	60	Case reports; 6-week period intervention	Twelve sessions were completed during the experimental period. The plyometric training program consisted of the following exercises: hurdle jumps, depth jumps, lateral jumps over box jumps, lunge jumps and vertical jumps. The number of sets per session varied between 2 and 4, and the repetitions between 6 and 12.	Meaningful within-plyometric group improvements were observed in right (+2.36 cm) and left (+2.48 cm) foot block jump, crossover jump (+2.64 cm) and sidestep block jump (+3.36 cm) performance. Changes were also significantly different from the control group.
Çankaya, Arabacı [51]	Women	16 yo	10	Case reports; 4-week period intervention	Six experimental sessions per week. Three sets of 30 jumps were added to the regular training session.	Meaningful increases in jumping height were found between the baseline (33.8 cm) and weeks 3 (36.0 cm) and 4 (36.4 cm).
Ramlan, Pitil [50]	Men and women	21 yo	12	Randomized controlled trial; 4-week period intervention	Two experimental groups: Plyometrics on grass and plyometrics on concrete surface. Both groups trained twice a week with the same number of sets, repetitions and resting time. The programs included the following exercises: drop from a platform, double leg jump over a hurdle, double leg drop jump and double leg drop jump over a hurdle. The number of repetitions varied between 3 and 6 sets of 12 and 7 repetitions, depending on the exercise.	Both groups improved their squat and counter-movement jumps after the training period. The plyometrics group on grass improved from 38.83 to 42.00 cm in the squat jump, and the plyometrics group on concrete from 39.33 to 41.50 cm. Moreover, the plyometrics group on grass improved their counter-movement jumps from 36.83 to 40.17 cm, and the plyometrics group on concrete from 35.33 to 39.00 cm. No significant differences between groups were found.

Yo: years old; VJ: vertical jump; N.R.: not reported.

**Table 3 ijerph-16-02960-t003:** Studies that investigated the effects of plyometric training programs on strength performance.

Study	Sex	Age (Mean) and Competitive Level	*n*	Design	Training Protocol	Main Results
Martel, Harmer [42]	Women	Aquatic group: 14 yo control group: 15 yo	19	Randomized controlled trial; 6-week period intervention	Aquatic plyometric training (APT): 2 × week; 45 min. Power skips, spike approaches, single- and double-leg bounding, and squat jumps progressively increased from two times per session to five times per session. Bouts increased from 10 s to 30 s of maximal jump during the period. Depth jumps were performed involving three submerged boxes also progressively increasing the number of times session during the period. Control group (CG): A flexibility supervised program was conducted twice a week, consisting of three sets of 8–10 static stretches for 30 s each.	There were no significant differences in concentric peak torque in either the dominant or nondominant leg between the APT and traditional volleyball training groups at baseline. Similar significant improvements in concentric peak torque were observed in the dominant leg of both groups when comparing baseline values with those obtained after 6 weeks. The improvements in both groups were similar for knee extension and flexion at both 60º and 180º.
Hewett, Stroupe [44]	Women	15 yo	20	Case reports; 6-week period intervention	Experimental group had three sessions per week. The program followed three phases: technique phase (two first weeks), fundamentals phase (using a proper technique to build strength and power), and performance phase (focusing on achieving maximal jumping).	Isokinetic peak torque increased 26% in the non-dominant leg and 13% in the dominant leg. The hamstring-to-quadriceps muscle peak torque ratio increased 13% on the dominant side and 26% on the non-dominant side.
Mroczek, Superlak [53]	Men	21 yo 2nd league	16	Case reports; 6-week period intervention	Measuring muscle stiffness: Three measurements performed once per week over 6 consecutive weeks of plyometric training (before the warm-up): In week 0, week 4 (the effects of the training completed in week 3) and week 6 (the effects of the training carried out in week 5). Individual assessments lasted up to 4 min, and the participants underwent them in a random order.	The analysis of stiffness levels in the posterior parts of the thigh revealed significant differences between the points in the left and right limbs only in the posterior muscles. Significant differences were observed for the semitendinosus immediately before the experiment started, whereas the differences were insignificant in the fourth and sixth training sessions.
Bashir, SulehHayyat [55]	Men	N. R.	45 G1—plyometric training (15); G2—weight and plyometric (15); G3—control group (15)	Randomized controlled trial; 12-week period intervention	Group I and II underwent respective training programs for 3 days per week for 12 weeks under the instruction and supervision of the investigator. Group-I performed plyometric training with a training intensity of 65%–80% of their 1RM and the subjects of experimental Group-II performed a combination of weight and plyometric training with a training intensity of 65%–80% of their 1RM.	Differences in muscular strength between plyometric training and control groups were significant at the 0.05 level of confidence. No significant difference between plyometric and combination of weight and plyometric training groups (0.37) in muscular strength after the training program. Differences in muscular endurance between plyometric training and control groups and a combination of weight and plyometric training and control group were significant. No significant difference between plyometric and combination of weight and plyometric training groups on muscular endurance after the training program.

Yo: years old.

**Table 4 ijerph-16-02960-t004:** Studies that investigated the effects of plyometric training programs on horizontal jump performance.

Study	Sex	Age (Mean) and Competitive Level	*n*	Design	Training Protocol	Main Results
Milić, Nejić [52]	N.R.	16 yo	46	Case reports; 6-week period intervention	Plyometric training group: 2 to 3 times per week (15 sessions in total during the experimental period). Five exercises (hurdle jump, depth jump, box jump sideways, lunge jumps and vertical jumps) for explosive leg strength. The number of sets varied between 2 and 4, and the repetitions between 6 and 12 during the training period.	A considerable increase in jumping skill was found among the members of the experimental group. Regarding the standing long jump, results reveal significant values (F = 5.55; *p* = 0.024).
Vassil and Bazanovk [54]	Men and women	14.4 for women and 17.0 yo for men	21	Case reports; 16-week period intervention	Twice a week. Squat jumps, lateral box push-offs, overhead throws, split squats, power drop, depth jumps, lateral hurdle jumps, plyometric push-ups and single-leg lateral hops were implemented. Each session had six exercises repeated twice (two sets), varying the repetitions between 8 and 15.	The women averaged changes from 194.8 ± 13.2 cm to 203.3 ± 13.2 cm (*p* > 0.05). and men’s results averaged improvements of 240.9 ± 16.7 cm to 248 ± 15.5 cm (*p* > 0.05). The women’s average depth leap long jump girl’s group average increased from 185.3 ± 14.7 cm to 193.8 ± 13.6 cm (*p* > 0.05), and the men’s results averaged an increase from 238.3 ± 17 cm to 246.4 ± 17.7 cm (*p* > 0.05).
Gjinovci, Idrizovic [48]	Women	21.9 yo	41	Randomized controlled trial; 12-week period intervention	Two experimental groups: Plyometric and skill-based. Each group had two sessions per week. Plyometric training included lower-body exercises (leg hops, vertical jumps, tuck jumps, lateral/diagonal jumps, broad jumps, obstacle jumps, box jumps and drop jumps) and upper body exercises (throwing exercises). The total of sets/week varied between 12 and 24 depending on the body part, and the repetitions between 40 and 58/week. Skill-based training consisted of volleyball drills, small-sided games and game drills.	The plyometric group significantly (*p* < 0.05) reduced their body-mass (trivial ES differences; 1% pre- to post-measurement changes) and improved their performance in the horizontal jump test (moderate ES differences; 7.6% changes). Players involved in skill-based-conditioning improved their capacities for horizontal jumping (small ES differences; 3.1% changes).
Çımenlı, Koç [57]	Male	18 to 24 yo	*N* = 36 12 control group, 12 wooden surface group, 12 synthetic surface group	Randomized controlled trial; 12-week period intervention	Plyometric training was practiced 3 days per week for 8 weeks. Each training session lasted about 50–60 min. Subjects performed 1 or 2 sets of 10 repetitions according to the training number (1 to 24). The tests applied to verify the horizontal jump were the right and left foot jump; double foot jump; right and left foot by taking a step.	In intra-group comparisons of the control group’s right foot, left foot, double foot and left foot by taking one step jump and the experimental group’s right foot, left foot, double foot, right foot by taking one step jump and left foot by taking one step jump values displayed a significant difference (*p* < 0.05). However, the experimental group’s right foot taking one step jump values did not differ significantly from the control group’s. During the post-test comparisons between groups, a significant difference was found between the right foot, taking a step with the right foot, and taking a step with the left foot values (*p* < 0.05). On the other hand, the right foot and double foot jump values did not differ significantly.

Yo: years old; N.R.: not reported.

**Table 5 ijerph-16-02960-t005:** Studies that investigated the effects of plyometric training programs on flexibility.

Study	Sex	Age (Mean) and Competitive Level	*n*	Design	Training Protocol	Main Results
Idrizovic, Gjinovci [56]	Women	16.6 yo	47	Randomized controlled trial; 12-week period intervention	Three groups: Plyometric, skill-based and control. The plyometric and skill-based groups had two sessions per week in addition to their regular training. The plyometric training lasted 10–15 min/session, and the skill-based training lasted 20–30 min. The plyometric training consisted of upper and lower limb exercises. The sets of plyometrics per training varied between 5 and 6, and the repetitions between 1 and 5. Rest between sets varied between 2 and 5 min. The skill-based training consisted of volleyball drills, small-sided games and real-games drills.	The main significant analysis of variance effects for time was observed for SIT-AND- REACH (F = 75.93, *p* < 0.01; small ES). Significant group × time interactions were observed SIT-AND-REACH (F = 11.70, *p* < 0.01; large ES). Post hoc differences were significant for SIT-AND-REACH, with better training effects of plyometric and skill-based conditioning when compared with the control program (9.1%; almost certainly positive).
Turgut, Colakoglu [43]	Women	Weighted jump rope group: 15.0 yo; standard jump rope group 14.1 yo; Control group: 14.4 yo	25	Randomized controlled trial; 12 week period intervention	Group I—Weighted jump rope training group: Performed rope jumping with weighted ropes and followed the program for twelve weeks, three times weekly. Group II—Standard jump rope training group: Followed the program for twelve weeks, three times weekly. Control Group: Followed a routine volleyball training program. Anaerobic power was measured by a vertical jump test (Lewis formula: Power = √4.9 x body mass (kg) × √vertical jump score (m) × 9.81); 30 m sprint test; hexagonal obstacle test and zigzag test; sit and reach test.	There was a statistically significant main effect of time (F = 59.05; *p* < 0.001) for sit and reach test outcomes (24.9 cm for recordings at baseline versus 28.5 cm for recordings after 12 weeks of training), indicating that all groups gained flexibility according to sit and reach test results.

Yo: years old.

**Table 6 ijerph-16-02960-t006:** Studies that investigated the effects of plyometric training programs on agility/speed.

Study	Sex	Age (Mean) and Competitive Level	*n*	Design	Training Protocol	Main Results
Lehnert, Lamrová [46]	Women	14.8 yo	11	Case Report; 8-week period intervention	Tests before and after plyometric sessions: Standing vertical jump (height of the jump in cm), vertical jump with approach (height of the jump in cm), shuttle run for 6 × 6 m 2 × week The training program was divided into three cycles. The number of sets varied between 3 and 4, and the repetitions between 8 and 10. No information about the resting period was provided.	Positive trend with differences (with no significant values—Z = 3.01) between speed values during the training program.
Idrizovic, Gjinovci [56]	Women	16.6 yo	47	Randomized controlled trial; 12-week period intervention	Three groups: Plyometric, skill-based and control. The plyometric and skill-based groups had two sessions per week in addition to their regular training. The plyometric training lasted 10–15 min/session, and the skill-based training lasted 20–30 min. The plyometric training consisted of upper and lower limb exercises. The sets of plyometrics per training varied between 5 and 6, and the repetitions between 1 and 5. Rest between sets varied between 2 and 5 min. The skill-based training consisted of volleyball drills, small-sided games and real-games drills.	The main effects for groups were significant for SPRINT20M (F = 3.77, *p* < 0.05; large ES). Post hoc analyses indicated greater effects of plyometric training in comparison with the other two training programs for SPRINT20M.
Sheikh and Hassan [59]	Male	Between 18 and 22 yo	*N* = 45 Experimental Group: I and II (15 + 15) Control Group: 15	Randomized controlled trial; 12-week period intervention	Group I and II—12 week, 3 × week (45 min per session) Exercises: 50 m sprint; shuttle run; side to side leg bounding, jump to box; tuck jump; depth jump. Group I: Plyometric training with weighted vest (2 kg) Group II: Plyometric training without weighted vest Side to side leg bounding, jump to box, tuck jump, depth jump. These exercises were performed for 45 min each day.	There is a significant difference between the plyometric training with a weighted vest group and the control group as well as between the plyometric training without a weighted vest group and the control group in terms of agility. Twelve weeks of plyometric training with a weighted vest resulted in greater improvements than twelve weeks of plyometric training without a weighted.

Yo: years old

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
