# Peer review of "The Effect of Plyometric Training in Volleyball Players: A Systematic Review"

_ijerph, 2019, doi:10.3390/ijerph16162960_

Round 1

Reviewer 1 Report

1.       Abstract:  

The authors have reported following PRISMA recommendations to conduct the systematic review. Please be sure to follow all sections suggested by PRISMA, whenever possible:

“Provide a structured summary including, as applicable: background; objectives; data sources; study eligibility criteria, participants, and interventions; study appraisal and synthesis methods; results; limitations; conclusions and implications of key findings; systematic review registration number”.

2.       Introduction:

 Pag. 2, line 28:

The main difference between your work and Stojanović et al. 2017 is that in your study you analysed (qualitatively) the factors sex and age. Therefore, the authors should explain deeply how these factors might affect the adaptations which plyometric training produces. In addition, the objectives should meet PRISMA's recommendations:

Provide a structured summary including, as applicable: background; objectives; data sources; study eligibility criteria, participants, and interventions; study appraisal and synthesis methods; results; limitations; conclusions and implications of key findings; systematic review registration number”.

 3.       Materials and methods:

Has the search protocol been registered? if yes, you should include it in the abstract and materials and methods

The information is confused and not well understood. Please, you should split the methodology into the following sections:

1)      Database sources and searches

2)      Inclusion and exclusion criteria: What type of studies have you selected?

3)      Outcomes

4)      Study screening and selection

5)      Data extraction and risk of bias assessment (study quality)

Figure 1 should be included in results and not methodology (section 3.1). Please, you should complete all boxes with the number of studies included.   

References are incorrectly cited in all the tables. For example, the reference 41 (Martel et al.) actually is the reference 31 (References section).

Please, attach the STROBE scale and the score for each study as a supplementary table.

 4.       Results:

The studies have not been correctly classified, for example Milic's study is not a cohort study. Furthermore, I consider that none of the included studies are cohort studies. I suggest to the authors to carry out a review of these designs. What method have you used to classify the studies?

You should include a specific section with the results of Risk of Bias.

Please, remove the duplicate sections of the tables (for example in the table 1, the study 41 has exactly the same information to table 2, except the "main results" column. This additional information is not enough for justifying the inclusion of a new table.

 5.       Discussion:

Sentence (pag. 16, line 22-23): “…the analyzed studies showed that plyometric training improved strength performance in the dominant leg of participants [41]”. I don't know if it refers to the correct reference (there are several errors in the citations). In addition, you use the plural and only one article is cited.

The authors should discuss in greater depth each section of the discussion specifically on the aspects (age and sex) indicated in objective 2.

Author Response

Reviewer 1

Abstract:  

The authors have reported following PRISMA recommendations to conduct the systematic review. Please be sure to follow all sections suggested by PRISMA, whenever possible:

“Provide a structured summary including, as applicable: background; objectives; data sources; study eligibility criteria, participants, and interventions; study appraisal and synthesis methods; results; limitations; conclusions and implications of key findings; systematic review registration number”.

Authors: The authors acknowledge the indications to improve the abstract. Changes were done in this section to answer the PRISMA recommendations.

Introduction:

Pag. 2, line 28: 

The main difference between your work and Stojanović et al. 2017 is that in your study you analysed (qualitatively) the factors sex and age. Therefore, the authors should explain deeply how these factors might affect the adaptations which plyometric training produces. In addition, the objectives should meet PRISMA's recommendations:

“Provide a structured summary including, as applicable: background; objectives; data sources; study eligibility criteria, participants, and interventions; study appraisal and synthesis methods; results; limitations; conclusions and implications of key findings; systematic review registration number”. 

Authors: Thank you for your suggestion. It was added in the text some differences between males and females that could affect the jumping performance.

Materials and methods:

Has the search protocol been registered? if yes, you should include it in the abstract and materials and methods 

Authors: Dear reviewer, thank you for your advice but, in fact, the present search protocol was not registered.

The information is confused and not well understood. Please, you should split the methodology into the following sections:

1)      Database sources and searches

2)      Inclusion and exclusion criteria: What type of studies have you selected? 

3)      Outcomes

4)      Study screening and selection 

5)      Data extraction and risk of bias assessment (study quality) 

Authors: Dear reviewer, thank you for your suggestion. Changes have been made according to your suggestion. We did not FOLLOW the purpose of the SECTIONS; however we have tried to add more information regarding each topic in our methods section.

Figure 1 should be included in results and not methodology (section 3.1). Please, you should complete all boxes with the number of studies included.   

Authors: Dear reviewer, thank you for your suggestion. Changes have been made according to your suggestion.

References are incorrectly cited in all the tables. For example, the reference 41 (Martel et al.) actually is the reference 31 (References section).

Authors: Dear reviewer, thank you for your notice that. Changes have been made accordingly.

Please, attach the STROBE scale and the score for each study as a supplementary table. 

Authors: Dear reviewer, thank you for your suggestion. Although the explanation of STROBE scale is already described in the methodology, that scale was attached as an appendix.

Results:

The studies have not been correctly classified, for example Milic's study is not a cohort study. Furthermore, I consider that none of the included studies are cohort studies. I suggest to the authors to carry out a review of these designs. What method have you used to classify the studies?

You should include a specific section with the results of Risk of Bias.

Authors: DEAR REVIEWER, THANK YOU. WE HAVE DOUBLE-CHECKED THE DESIGNS.

Please, remove the duplicate sections of the tables (for example in the table 1, the study 41 has exactly the same information to table 2, except the "main results" column. This additional information is not enough for justifying the inclusion of a new table.

Authors: Dear reviewer, the authors understand your concerns, however each table expose the results for the different abilities. In fact, there are some studies that combined different abilities, this is the reason why the results are different in each table. Therefore, the authors understand that this information should be kept as it was not duplicated.

Discussion:

Sentence (pag. 16, line 22-23): “…the analyzed studies showed that plyometric training improved strength performance in the dominant leg of participants [41]”. I don't know if it refers to the correct reference (there are several errors in the citations). In addition, you use the plural and only one article is cited.

The authors should discuss in greater depth each section of the discussion specifically on the aspects (age and sex) indicated in objective 2.

Authors: Dear reviewer, thank you. We have updated the discussion, namely including the effects in both sexes and ages.

Reviewer 2 Report

General Comments

General Weaknesses 

Discussion

- In spite of Discussion is interesting, it expresses several significant deficits. Authors must present the possible underlying mechanisms of results of each parameter under study (vertical jump, strength, horizontal jump, flexibility and agility/speed) and they must also identify the literature lacks about this interesting topic for training and researching (after analyzing the meta-analysis results) to “guide the way” to a future research projects. Additionally, Authors must include the research project´s limitations at the end of this section (pages 15-18 of 22). 

Conclusions

- Authors must follow the journal guidelines to improve Conclusions (it results too general at the moment, Authors must try get specific conclusions considering the meta-analysis´ goals) (page 18 of 22).

General Strengths

- Introduction is correct and interesting (rationale is robust and it´s supported on enough and updated quotations) (pages 1-2 of 22). 

- This section is (Results) very interesting. It contents abundant useful data for readers and researchers (pages 4-14 22). This is one of the most important strengths of the manuscript.

Major Comments:

Title: 

Strengths

- Title is correct (page 1 of 22).

Abstract

Weaknesses 

- Abstract must be elaborated again. It must follow recommendations of the present report (page 1 of 22). 

Keywords

Weaknesses

- Keywords must be corrected. Authors must avoid use the same words in Title and Keywords (page 1 of 22).

Introduction

Strengths

- Introduction is correct and interesting (rationale is robust and it´s supported on enough and updated quotations) (pages 1-2 of 22). 

Methods

Weaknesses 

- Format mistake: Authors must write this term with capital letters (only the first word): The process for screening articles followed  the Preferred (R)reporting (I)items for (S)systematic (R)reviews and (M)metaanalyses (PRISMA) guidelines [40] (lines 1-2, page 3 of 22). 

- Authors write: “The abstracts of all the articles found were screened against the predefined selection criteria by the three authors of the present” (lines 7-8, page 3 of 22). How many Authors do sign this manuscript?, three?, six?... Authors must make clear this issue in the whole manuscript, since they state that two people “extrated data”… Thus, I recommend to avoid these kind of aspects on the manuscript (the Authors´ role must not be specified in the manuscript). 

- Flowchart: Figure 1 demands improvements. Lateral boxes unreadable (page 3 of 22). 

- I recommend to use italics for these kind of terms: (e.g., i.e., et al.,), I mean: (e.g., i.e., et al.,). 

- Authors must not write any name in the manuscript: “Afterwards, two independent reviewers (Filipe Clemente and Ricardo Lima)” (line 9, page 4 of 22).

Results

Weaknesses 

- Format mistake: Authors must write this expression in italics: post hoc (line 2 page 14 of 22) since they are using Latin language in an English text. 

Strengths

- This section is (Results) very interesting. It contents abundant useful data for readers and researchers (pages 4-14 22). This is one of the most important strengths of the manuscript.

Discussion

Weaknesses

- In spite of Discussion is interesting, it expresses several significant deficits. Authors must present the possible underlying mechanisms of results of each parameter under study (vertical jump, strength, horizontal jump, flexibility and agility/speed) and they must also identify the literature lacks about this interesting topic for training and researching (after analyzing the meta-analysis results) to “guide the way” to a future research projects. Additionally, Authors must include the research project´s limitations at the end of this section (pages 15-18 of 22). 

Conclusions

Weaknesses

- Authors must follow the journal guidelines to improve Conclusions (it results too general at the moment, Authors must try get specific conclusions considering the meta-analysis´ goals) (page 18 of 22).

References

- This section must be checked it in details. It could contain format mistakes (pages 18 of 22).    

Tables and Figures

Weaknesses

- Flowchart: Figure 1 demands improvements. Lateral boxes unreadable (page 3 of 22). 

Strengths

- Tables 1, 2, 3, 4 and 5 are very interesting. It present a lot of interesting and useful data. 

Author Response

General Comments

General Weaknesses 

Discussion

- In spite of Discussion is interesting, it expresses several significant deficits. Authors must present the possible underlying mechanisms of results of each parameter under study (vertical jump, strength, horizontal jump, flexibility and agility/speed) and they must also identify the literature lacks about this interesting topic for training and researching (after analyzing the meta-analysis results) to “guide the way” to a future research projects. Additionally, Authors must include the research project´s limitations at the end of this section (pages 15-18 of 22). 

Authors: Dear Reviewer, thank you. We have updated the discussion aiming to increase the mechanisms that explains the outcomes and also adding some recommendations for future studies in each area.

Conclusions

- Authors must follow the journal guidelines to improve Conclusions (it results too general at the moment, Authors must try get specific conclusions considering the meta-analysis´ goals) (page 18 of 22).

Authors: Dear reviewer, thank you. We have updated the conclusion session.

General Strengths

- Introduction is correct and interesting (rationale is robust and it´s supported on enough and updated quotations) (pages 1-2 of 22). 

Authors: The authors acknowledge the reviewer’s words.

- This section is (Results) very interesting. It contents abundant useful data for readers and researchers (pages 4-14 22). This is one of the most important strengths of the manuscript.

Authors: Dear reviewer, thank you for your comment.

Major Comments:

Title: 

Strengths

- Title is correct (page 1 of 22).

Authors: The authors acknowledge the reviewer’s words.

Abstract

Weaknesses 

- Abstract must be elaborated again. It must follow recommendations of the present report (page 1 of 22). 

Authors: Changes were done to improve this section. Thank you for your suggestion.

Keywords

Weaknesses

- Keywords must be corrected. Authors must avoid use the same words in Title and Keywords (page 1 of 22).

Authors: It was changed as suggested.

Introduction

Strengths

Authors: The authors acknowledge the reviewer’s words.

- Introduction is correct and interesting (rationale is robust and it´s supported on enough and updated quotations) (pages 1-2 of 22). 

Authors: The authors acknowledge the reviewer’s words.

Methods

Weaknesses 

- Format mistake: Authors must write this term with capital letters (only the first word): The process for screening articles followed  the Preferred (R)reporting (I)items for (S)systematic (R)reviews and (M)meta‐analyses (PRISMA) guidelines [40] (lines 1-2, page 3 of 22). 

Authors: Dear reviewer, thank you. We have changed accordingly.

- Authors write: “The abstracts of all the articles found were screened against the predefined selection criteria by the three authors of the present” (lines 7-8, page 3 of 22). How many Authors do sign this manuscript?, three?, six?... Authors must make clear this issue in the whole manuscript, since they state that two people “extrated data”… Thus, I recommend to avoid these kind of aspects on the manuscript (the Authors´ role must not be specified in the manuscript). 

Authors: Dear reviewer, thank you. We have changed accordingly.

- Flowchart: Figure 1 demands improvements. Lateral boxes unreadable (page 3 of 22). 

Authors: Dear reviewer, thank you. We have changed the figure.

- I recommend to use italics for these kind of terms: (e.g., i.e., et al.,), I mean: (e.g., i.e., et al.,). 

Authors: Dear reviewer, thank you. We have changed accordingly.

- Authors must not write any name in the manuscript: “Afterwards, two independent reviewers (Filipe Clemente and Ricardo Lima)” (line 9, page 4 of 22).

Authors: Dear reviewer, thank you. We have removed.

Results

Weaknesses 

- Format mistake: Authors must write this expression in italics: post hoc (line 2 page 14 of 22) since they are using Latin language in an English text. 

Authors: Dear reviewer, thank you. We have changed accordingly.

Strengths

- This section is (Results) very interesting. It contents abundant useful data for readers and researchers (pages 4-14 22). This is one of the most important strengths of the manuscript.

Authors: Dear reviewer, thank you for your comment.

Discussion

Weaknesses

- In spite of Discussion is interesting, it expresses several significant deficits. Authors must present the possible underlying mechanisms of results of each parameter under study (vertical jump, strength, horizontal jump, flexibility and agility/speed) and they must also identify the literature lacks about this interesting topic for training and researching (after analyzing the meta-analysis results) to “guide the way” to a future research projects. Additionally, Authors must include the research project´s limitations at the end of this section (pages 15-18 of 22). 

Authors: Dear reviewer, thank you. We have updated the discussion aiming to increase the mechanisms that explains the outcomes and also adding some recommendations for future studies in each area.

Conclusions

Weaknesses

- Authors must follow the journal guidelines to improve Conclusions (it results too general at the moment, Authors must try get specific conclusions considering the meta-analysis´ goals) (page 18 of 22).

Authors: The authors tried to resume better the study in this session, explaining the results found.

References

- This section must be checked it in detail. It could contain format mistakes (pages 18 of 22).    

Authors: Dear reviewer, thank you. We have downloaded the EndNote file for references in the journal site and rectified the references.

Tables and Figures

Weaknesses

- Flowchart: Figure 1 demands improvements. Lateral boxes unreadable (page 3 of 22). 

Authors: Dear reviewer, thank you. We have changed accordingly with your suggestions.

Strengths

- Tables 1, 2, 3, 4 and 5 are very interesting. It present a lot of interesting and useful data. 

Authors: Dear reviewer, thank you for your comment.

Round 2

Reviewer 1 Report

The authors have done the suggested changes in the previous revision and I would like to thank them for considering the opinion of this reviewer and modifiying the manuscript accordingly. The only point I do not agree with is how the classification of the studies for the systematic review was performed. The studies shouldn't be classified as RCT or case report.

Reviewer 2 Report

Manuscript has been improved. From my point of view, it might make a worthwhile contribution to knowledge.